

# GC Insights: The crystal structures behind the optical properties of minerals – a case study of using TotBlocks in an undergraduate mineralogy lab

Derek D.V. Leung[1,2], Paige E. dePolo[1]

[1]School of GeoSciences, The University of Edinburgh, Edinburgh, United Kingdom

[2]Harquail School of Earth Sciences, Laurentian University, Sudbury, Ontario, Canada

*Correspondence to:* Derek D.V. Leung (dleung@laurentian.ca)

**Abstract.** Spatial thinking represents an on-going challenge in geoscience education, but concrete manipulatives can bridge the gap by illustrating abstract concepts. In an undergraduate optical mineralogy lab session, TotBlocks were used to illustrate how mineral structures influence optical properties such as cleavage and pleochroism. More abstracted properties, e.g., extinction angles, were increasingly difficult to illustrate using this tool.

## 1 Introduction

Spatial thinking and understanding complex 3D structures mark fundamental challenges in geology education (Ishikawa and Kastens, 2005; Liben and Titus, 2012; Woods et al., 2016). These challenges extend to the atomic scale where the crystal structures of minerals are difficult to conceptualize (Dyar et al., 2004). Understanding crystal structures is important because the identifiable features of minerals – e.g., cleavage and pleochroism – ultimately arise from crystal structures and their inherent symmetry (Neumann, 1885). Thus, a more intuitive understanding of these abstract systems is desirable.

Current teaching strategies for visualizing crystal structures include physical manipulatives, e.g., ball-and-stick models and paper polyhedral models (Rodenbough et al., 2015; Wood et al., 2017; He et al., 1990a; 1990b; 1994) and virtual manipulatives, e.g., visualization software (Moyer et al., 2002; Extremera et al., 2020). 3D-printed physical manipulatives can illustrate unit cells in crystallography (Rodenbough et al., 2015), complex structures like DNA (Jittivadhna et al., 2010; Howell et al., 2019), and other chemical principles (Witzel, 2002; Kaliakin et al., 2015; Melaku et al., 2016; Smiar and Mendez, 2016; Geyer, 2017; Lesuer, 2019; Horikoshi, 2020; Melaku and Dabke, 2021).

The TotBlocks project aims to communicate the crystal structures of modular rock-forming chain and sheet silicate minerals (pyroxenes, amphiboles, micas, and clay minerals) through 3D-printed building blocks (Leung and dePolo, 2022a; Fig. 1a). This work investigates the utility of TotBlocks in communicating the relationship between the crystal structures and optical properties of minerals.





**Figure 1 (a) The crystal structure of the mica group, illustrated using TotBlocks (Leung and dePolo, 2022a). (b) Example of optical properties visible under the microscope. Biotite (mica group) displays a perfect basal cleavage on the {001} and displays the strongest pleochroic colour when the substage polarizer is parallel to the layers of octahedral modules in Fig. 1a (top image). (c) Respondents' understanding of concepts decreased with increasing abstractness. (d) Proposed spiral learning model for optical mineralogy, based on insight from Fig. 1c.**



## 2 Materials, Methods, Ethics Approval

A one-hour exercise on modular mineralogy (File S1 in the Supplement) was conducted during the last lab (April 2022) of a second-year Optical Mineralogy class at Laurentian University (Sudbury, Canada). After a brief introductory lecture, students sequentially built the crystal structures of the mica, pyroxene, and amphibole (super-)groups using TotBlocks. Using these models, students reflected on the optical properties (pleochroism, cleavage, and extinction angles) they had previously discussed during the semester (Fig. 1b). This session was voluntary for students and attendance was not monitored.

At the end of the exercise, an optional, anonymous feedback survey consisting of four Likert-scale questions and four free-response questions was distributed to the students (File S2 in the Supplement). Students self-assessed whether their understanding of optical properties was improved by the lab. They also reflected on what aspects of the lab worked well for them or could be improved. The data analyzed here (File S3 in the Supplement) were originally collected as teaching feedback. Ethical approval for secondary data usage was granted by the Laurentian University Research Ethics Board (LUREB; #6021264).

## 3 Results

Fifteen survey responses were collected. Within these surveys, two respondents (13 %) did not complete the self-assessment section and are tabulated as "no response" for all Likert-scale questions.

No respondents reported a "worse" understanding of topics at the end of the lab for any Likert-scale question (Fig. 1c). 87 % (13/15) of respondents reported that their understanding of modular mineralogy was "better" at the end of the lab and no respondents reported the "same" level of understanding. The survey responses for understanding pleochroism and cleavage angles were identical with 67 % (10/15) of respondents reporting they understood the concepts "better" and 20 % (3/15) reporting the "same" level of understanding. The survey responses for understanding of extinction angles were split more evenly with 47 % (7/15) of respondents reporting they understood the concept "better" and 40 % (6/15) reporting the "same" level of understanding. Excluding the two "no response" respondents, 100 % of respondents reported a "better" understanding of modular mineralogy, 77 % reported a "better" understanding of cleavage and pleochroism, and 54 % reported a "better" understanding of extinction angles (Fig. 1c).

All survey participants engaged with the free-response questions with a general positive consensus observed. Students reported impressions like they "enjoyed the experience" and that "the instructions were clear and the activity very dynamic."

## 4 Discussion

The use of TotBlocks in this lab setting allowed students to learn mineralogical concepts in alignment with the theory of experiential learning (sensu Kolb and Fry, 1975). Kolb and Fry (1975) conceptualize learning as an iterative, four-stage process that cycles through (1) concrete experience, (2) observations and reflections based upon that experience, (3) analysis



of those observations to form abstract conceptualizations, and (4) applying these conceptualizations to new experiences.
Through (1) the concrete experience of constructing a mineral structure with TotBocks, students engage in active and
cooperative learning (Smith et al., 2005), and (2) are invited to observe the modularity of different silicate minerals and
reflect on their structural relationships. These reflections provide (3) the abstract foundation for students to then (4) extend
these ideas to the physical properties of minerals and more complex aspects of crystal chemistry. The process of students
using physical manipulatives to solidify their understanding of crystal structures aligns TotBlocks with the educational
theory of constructionism (Harel and Papert, 1991).

The structure of the lab exercise additionally followed ideas of spiral learning for mineralogy teaching (Bruner, 1966; Dyar
et al., 2004). Students began with the mica structure – the protostructure for other modular rock-forming minerals – and
then were invited to actively build new concepts of this existing knowledge. Additional concepts of cleavage, pleochroism,
and extinction angles were introduced in context of the previously developed ideas and built upon the principles the students
had encountered. In essence, students began with chemical building blocks, progressed to crystal structures, and then
developed further understanding of optical properties (Fig. 1d).

Using TotBlocks to illustrate optical mineralogy principles in this classroom setting resulted in some preliminary successes.
Students felt the advantages of using physical manipulatives. One student noted "paralleling real-life structures into models"
was "easy to understand" while another reported "that seeing cleavage and extinction in real life" was an aspect of the lab
that worked well. Another student observed that "building" was "different in understanding than just being lectured." These
reported experiences illustrate the efficacy of TotBlocks for concretizing abstract ideas of crystal structures for students
similar to the pattern observed by Fencl and Heunink (2007) in physics classrooms. TotBlocks also allowed students to
productively engage in informal cooperative learning (Smith et al., 2005). A student reflected that "having to build the
structures as a group of 3-4 people really helped to share concepts and opinions about the question[s]." This experience
illustrates that the use of these manipulatives in the classroom can support peer-to-peer exchange of insights (Boud, 2001;
Keerthirathne, 2020). These responses suggest that TotBlocks supported both experiential and cooperative learning in this
lab.

Despite these successes, we observed a decrease in the students' understanding of key optical mineralogy principles with
increasing orders of complexity (Fig. 1c). Although the students' understanding of modular mineralogy improved, fewer
students reported similar improvements to their understanding of cleavage and pleochroism. The most challenging concept
to impart was extinction angles. This decrease in understanding corresponds to increasing abstractness of concepts from
basic building blocks and crystal structures to polarized light and the optical indicatrix, consistent with a spiral learning
model (Fig. 1d).

We also encountered several practical limitations within the lab, with the most notable being the short time allotted to the
exercise. The time restriction was evident for the mineral that concluded the lab, the amphibole structure. Three students
noted that building the amphibole structure was confusing, suggesting that additional time on that exercise would have been
beneficial. A potential solution would be integrating TotBlocks into multiple lab sessions. Increasing students' exposure to



TotBlocks throughout an academic term would allow students to learn how TotBlocks work as physical manipulatives prior
to applying them to understanding optical properties. Additionally, several students noted a need for additional support with
the construction instructions of the mineral structures in the lab. They shared thoughts like "I think the building of the
structures would be easier with step by step image (Ikea furniture)" and "it would be helpful to have step-by-step
instructions with images." These reflections demonstrate a need for more clarity in task presentation for students
(Rosenshine and Stevens, 1986; Rink, 1994). In future classroom applications of TotBlocks, additional building support
could be provided to the students through instructional videos (e.g. Leung and dePolo, 2022b). Finally, this study lacks a
control group. We do not know whether a student's experience of learning about modular mineralogy without the support
of TotBlocks would have been significantly better or worse.

Using TotBlocks as concrete manipulatives within experiential, spiral, and cooperative learning frameworks shows
potential for improving students understanding of optical mineralogy concepts. Incorporating TotBlocks with other
representations of crystal structure (e.g. ball-and-stick models and visualization software) in mineralogy classrooms merits
further study, particularly in the context of more extended use throughout a course (Tsui and Treagust, 2013).
**5 Data and Code Availability**
The full source code and 3D model files for the TotBlocks project (GPLv3 license) can be found on Github:
https://doi.org/10.5281/zenodo.5240816 (Leung, 2022).
**6 Supplement**
The supplement included in this contribution consists of three files: the lab manual (File S1), survey (File S2), and response
spreadsheet (File S3).
**7 Author Contributions**
DDVL conceptualized and designed TotBlocks, delivered the lab exercise, collated survey responses, and made the figure.
PEdP contextualized TotBlocks in the pedagogical literature and wrote the first draft of this manuscript. Both authors
designed the lab exercise and survey, and discussed and edited the manuscript.
**8 Acknowledgements**
We thank Sandra Hoy and Lise Carriere (LUREB) for consultation and assistance in submitting the ethics application and
two anonymous LUREB members for their comments strengthening the application. We thank Courtney Onstad (Simon
Fraser University) for her advice around the language used in ethics assessments. Andrew McDonald (Laurentian
University) provided access to the Optical Mineralogy lab session, and Christopher Beckett-Brown and Melissa Barerra
assisted. We thank Geoscience Communication executive editors Sam Illingworth and John Hillier for their advice and
guidance around this manuscript.



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
