# Peer review of "case study of using TotBlocks in an undergraduate optical mineralogy lab"

_EGUsphere, 2023_

## Referee Comment (RC1)

Review: Leung and de Polo, *The crystal structures behind the optical properties of minerals – a case study of using TotBlocks in an undergraduate mineralogy lab*

Comments

1. This is a new and innovative way to use 3D printed materials to support and enhance student learning. Having students build their own crystal models with these 3d printed components is really cool.
2. Application of learning theory to explicitly design and implement this teaching activity is laudable. This includes active learning, constructionalism, spiral learning trajectories, and cooperative learning. This is all really great.
3. Building the 3D structures of the biopyriboles is a great way for students to understand the relationship between crystal chemistry and crystal structure. Note that these approaches have been advocated since the mid 1990s, see two references below.
4. My main criticism is the extension of using these crystal structure models to demonstrate optical properties of minerals. Using the structures works moderately well for demonstrating the relation between crystal structure and physical properties like cleavage. However, the expression of optical properties such as pleochroism and extinction angle requires also that the orientation of principle vibration directions (X,Y,Z) are shown with respect to the crystallographic axes (a,b,c). This is not so difficult using biotite, or even ortho amphiboles and pyroxenes. But, this can be really confusing for the monoclinic varieties. I did not see that students were asked to locate the orientation of the vibration axes in or on the crystal structures. So I don't see how they can effectively determine what pleochroic colors should be attributed (usually denoted as e.g., X=green Y=brown Z= red), and similarly, I'm not sure how the extinction angle can be related to the crystal model without some representation of where the vibration axes are located.
5. For publication, I think the authors need to address the relationship between crystallographic orientation and orientation of vibration directions. These would be similar to the perspective crystal drawings showing crystal form, cleavage, crystal axes, and optical orientations in standard Mineralogy texts and atlases like Deer, Howie and Zussman. The models are really cool. Interpreting cleavage is a nice extension of the understanding of the crystal structures. But I think that extension to optical properties is a step too far without additional optical orientation information.

Review Criteria:
1. Addresses relevant scientific question. The challenge of developing spatial thinking by students is of interest and concern in most geoscience fields. This is particularly challenging when trying to teach Mineralogy and Crystallography.
2. A novel approach is used to create mineral models using 3D printing
3. The methods are clearly stated. Application of learning theory is a strength.
4. Results are clearly presented to show learning gains and limitations.
5. Prior work is cited appropriately. I provided two additional reference that might be included just for historical context.

6. The title accurately reflects the content of the article. Although I do question whether these models can be used to demonstrate optical properties of minerals without adding critical information about the orientation of principle vibration directions with respect to crystallographic axes.
7. The abstract is appropriate
8. The presentation is adequate, but some more description and context of what the students actually experienced in doing these activities would be interesting.
9. Language used is appropriat4e
10. References are appropriate.

Other Comments:

The use of manipulatives has been shown to strongly enhance learning about spatial objects—so this manuscript is well-grounded in established theory and practice. I think that this is a great exercise to demonstrate the fundamental architecture of the family of "biopyriboles" that have the same basic T-O-T layering but with single or double chains ot 2-D sheets. And it is a good way to demonstrate the physical property of cleavage, particularly for the perfect {100} cleavage of biotites. It's a bit of a jump for students to also see the {110} cleavage of pyroxenes and amphiboles, but in the correct orientation looking down [001] (i.e., the "end section view") students should be able to see the cleavage following the weaker bonds between the M sites at ~90 degrees for single chains and ~120-60 degrees for double chains. That's all great stuff. For the clinopyroxene and clinoamphibole examples, question a) what type of cleavage and angles, could use some more direction as only one cleavage direction will be seen looking perpendicular to any of the prismatic faces (parallel to c or [001]) and the two cleavages and their angle will only be observed if you know to look down the c-axis or [001]. It would be a better learning exercise if students were directed to look down different directions on the model and then compare features.

I have a little bit of an issue extending this application to optical properties as these have to be understood in terms of the relationship between the crystallographic axes (a,b,c) and the principal vibration directions (X,Y,Z). For pleochroism in biotite, this is easy to demonstrate as biotite is close to pseudo-hexagonal (monoclinic in detail), so pleochroism can easily be demonstrated parallel to the E-W or N-S cross hairs with e.g., X= brown and Y=Z dark brown. But for amphiboles and pyroxenes, this becomes more complex depending on a) is the crystal system orthorhombic or monoclinic, and b) the pleochroic color depends on which vibration direction you are looking down, and if you're not looking down one of the vibration axes you will get some intermediate color. Similarly, with extinction angle I can see why comprehension of the students did not improve as much. Again, with biotite it's simple to demonstrate parallel extinction with respect to {001} cleavage which is parallel with a vibration direction. This is also easy with the ortho amphiboles and pyroxenes where the vibration directions are parallel with crystallographic directions. But for the clino amphiboles and pyroxenes this is much more complicated for students. To be in the proper orientation to measure extinction angle, you have to be looking down the optic normal (Y) to be able to measure the correct Z to c angle. That requires understanding numerous rotational degrees of freedom to a get the crystallographic

axis aligned with a cross hair, and then b) longitudinally rotate the crystal so that the optic normal is parallel with the line of sight. Any other orientation will result in an incorrect extinction angle that can range from 0 degrees (parallel) to the actual extinction angle of ~20 degrees for many clinoamphiboles or ~40 degrees for many clinopyroxenes. So, I guess the shortfall that I see here is not being able to simultaneously show the vibration directions compared to the crystallographic axes in these ToTblock models.

I'm delighted to see the application of learning theory with reference to Kolb and Fry (1975) and constructionism (Harel and Papert, 1991) (I would use constructivist theory as an alternate expression, but a rose is a rose). "Spiral" learning progressions and cooperative learning approaches are also applied in this activity, and this is also supported by learning theory.

I don't think you can answer the question for clinopyroxene part b, which orientation would show inclined extinction without having reference to the vibration directions. And for part ii) I don't think the model as shown can address the question of what orientation is needed to demonstrate clino (inclined) from ortho (parallel) pyroxenes based on extinction angle.

Lines 20-21, In the early Teaching Mineralogy workbook that derived from a NSF-sponsored workshop followed by publication of lab exercises by MSA, two early contributions used this approach using traditional ball and stick models and also building crystal structures with templates using plasticene balls. The use of 3D printing of crystal structure components is a really nice natural evolution of this tradition.

Mogk, D.W., Directed-Discovery of Crystal Structures Using Ball and Stick Models, in Brady, J., Mogk, D. W., and Perkins, D., (editors), 1997, Teaching Mineralogy, a workbook published by the Mineralogical Society of America, 406 pp. available on line at
https://serc.carleton.edu/NAGTWorkshops/mineralogy/activities/ballstick.html

Hollacher, K., Building Crystal Structure Ball Models Using Pre-Drilled Templates: Sheet Structures, Tridymite, and Cristobalite, in Brady, J., Mogk, D. W., and Perkins, D., (editors), 1997, Teaching Mineralogy, a workbook published by the Mineralogical Society of America, 406 pp. available online at
https://serc.carleton.edu/NAGTWorkshops/mineralogy/activities/buildball.html

---

## Author Comment (AC1)

**Optical Mineralogy Lab:**
**Getting started with TotBlocks**

The biopyriboles: LEGO-like minerals

Last revised May 2, 2023

The building blocks of modular rock-forming minerals consist of tetrahedral (*T*) and octahedral (*O*) modules. There are two different variations of each module, reflecting the direction that these modules point. We denote these two variations with '+' and '−', with + meaning that the apices of the polyhedra are pointing upward when the widthwise pegs are pointing rightward, whereas − means that the apices of the polyhedral are pointing downward. Note that the *T* modules are reversible, so the + and − notation is relative and not restricted to the modules themselves. For the *T* modules, we'll use both the $T^+$ and $T^-$ variations. For the *O* modules, we will only use the $O^+$ modules (no $O^-$ modules).

[Figure]

These modules are connected to make *T-O-T* modules. **In this lab, we'll keep the *T-O-T* modules together (we won't break them apart)**, but instead we'll look at how *T-O-T* modules can be linked together widthwise.

To unlink the *T-O-T* modules, always pull apart parallel to the pegs. Do not twist the pegs or they will break! However, if by chance a module breaks, don't worry. 3D printing is cheap; broken modules can easily be replaced. Please tell a TA if a module is broken.

**The most important takeaway of this lab is to go through the experience of building crystal structures.** The questions in this lab manual are there solely to guide you to think about the structural relationships between different minerals, as well as the fractal relationship between crystal structures and physical/optical properties.

**Lab activity**

**Mica group**

1. In front of you is the assembled crystal structure of the mica group, structural formula $IM_3T_4O_{10}A_2$, which is a 2:1 phyllosilicate with interlayer cations. The main mica-group minerals are annite-phlogopite (biotite series), $K(Fe^{2+},Mg)_3(AlSi_3O_{10})(OH)_2$, and muscovite, $KAl_2(AlSi_3O_{10})(OH)_2$.

   The red layers represent infinite sheets of vertex-sharing tetrahedra ($T$), commonly occupied by Si and Al; whereas the cyan layers represent infinite sheets of edge-sharing octahedra ($O$), commonly occupied by Mg, Fe, and Al. Black vertical pegs are used to hold the sheets together. In the space between the $T$-$O$-$T$ modules is the interlayer ($I$), where large cations such as K, Na, and Ca reside (these are not shown by TotBlocks).

[Figure]

   a. Describe the cleavage of a mica (number of cleavages, angle). How does the crystal structure of the mica group dictate its cleavage? Mark the cleavage plane on the picture above.

   b. Annite (the Fe-bearing member of the biotite series) commonly displays strong pleochroism. Given the sheeted crystal structure of annite and where Fe is located within it, explain why annite is strongly pleochroic.

   c. Assuming that the substage polarizer is oriented N-S, when looking down (parallel to) the cleavage (see example photo above), is the absorption (*i.e.*, colour) stronger when the cleavage is oriented N-S or E-W?

      i. Check this on the microscope. Is this consistent with what you expected?

**Pyroxene group**

2. The pyroxenes are single-chain inosilicates with the structural formula **M2M1T$_2$O$_6$**. Common pyroxenes include the enstatite-ferrosilite series, $(Mg,Fe^{2+})_2Si_2O_6$, and diopside, $CaMgSi_2O_6$.

   Aside from the *T-O-T* modules (hosting the *T* and *M*1 sites, respectively) in the structure, there is another site that borders the *O* modules (the *M*2 site), which is slightly larger than the *M*1 site in the *O* module (not shown). We'll focus on the building the crystal structure of the clinopyroxenes (see figure below):

   • Connect one single-chain-width *T-O-T* module (labelled as ①) to a second single-chain-width module ② such that the *T⁻* module of the first ① is joined to the bottom of the *O* module of the second ② [and top of *O* of ① joined to *T⁺* of ②].

   • A third single-chain-module ③ can then be joined so that the *T⁺* of the first module ① is joined to the top of the *O* module of the third [and bottom of *O* of ① joined to the *T⁻* of ②].

   • A fourth *T-O-T* module ④ is joined to the second ② and third ③ *T-O-T* modules.

[Figure]

**Clinopyroxene (*C*2/*c* or *P*2$_1$/*c*)**

   a. Based on the assembled structure, what type of cleavage (number, angle) would a clinopyroxene display when viewed along the *c* axis ([001]), as shown on the figure above? Draw this on the figure.

[Figure]

Clinopyroxenes

[Figure]

Orthopyroxenes

Images from pp. 205 and 209 in Introduction to Optical Mineralogy by Nesse (1991).

b. Clinopyroxenes can be distinguished from orthopyroxenes by their extinction angles. The figures above show the orientations of the principal vibration directions (X, Y, and Z) with respect to the crystallographic axes (*a*, *b*, and *c*) for clinopyroxenes (top) and orthopyroxenes (bottom), along with the orientations of the two cleavage planes.

    i. Based on the assembled clinopyroxene structure and the figures above, which orientation (*i.e.*, looking down *a*, *b*, or *c*) would show inclined extinction? What kinds of extinction do the other two orientations show?

    ii. Why is this relevant when it comes to differentiating clinopyroxenes from orthopyroxenes? In other words, can you use any orientation to distinguish between clinopyroxenes and orthopyroxenes?

**Amphibole supergroup**

We will now **disassemble the mica structure** to demonstrate that the building blocks of the micas are the same as those found in the pyroxenes and amphiboles.

- Holding the mica structure sideways (one hand holding the top two *T-O-T* modules, the other holding the bottom two *T-O-T* modules), pull the two sets of modules apart to pull out the vertical pegs. **Do not twist the modules!**

- The grey vertical pegs will be stuck in one of the modules. Pull these pegs straight out of the modules.

- For each of the two sets of mica structures, hold one *T-O-T* module in each hand and **carefully** pull apart horizontally. **Do not twist the modules!** Repeat for the other set.

- You should now have four separate double-chain *T-O-T* modules and two vertical pegs. You won't need the vertical pegs anymore, so give these back to the TAs.

3. The amphiboles are double-chain (ino-)silicates with the general structural formula $AB_2C_5T_8O_{22}W_2$ and include the following minerals: cummingtonite-gruenerite series, $(Mg,Fe^{2+})_2(Mg,Fe^{2+})_5Si_8O_{22}(OH)_2$; hornblende series, $Ca_2(Mg,Fe^{2+},Al)_5(Si,Al)_8O_{22}(OH)_2$; and tremolite-actinolite series, $Ca_2(Mg,Fe^{2+})_5Si_8O_{22}(OH)_2$.

   Like the pyroxenes, the crystal structure of the amphiboles contains an extra site that borders the *O* modules called the *B* (or *M4*) site. Like the micas, there is also a site that lies in between the *T-O-T* modules (*A* site). We'll focus on constructing the clinoamphibole structure (see figure below):

- Connect one double-chain-width *T-O⁺-T* module ① to a second double-chain-width module ②, such that the *T⁻* module of ① is joined to the bottom of the *O* module of ② (the top of *O* for ① will also connect to the *T⁺* of ②).

- A third double-chain module ③ can then be joined so that the *T⁺* of the first module ① is joined to the top of the *O* module of the third ③ (the bottom of *O* for ① should also join to the *T⁻* of ③).

- A fourth module ④ is joined to the second ② and third ③ modules.

[Figure]

[Figure]

a. Now, let's consider the cleavages of the pyroxenes, amphiboles, and micas all together. You should have the assembled structures of the pyroxenes and amphiboles in front of you. For reference, there is an extra mica structure on display at the front of the classroom. The pyroxenes have the shortest module width (single chains), followed by the amphiboles (double chains), and the micas consist of sheets.

    i. As the module width increases from single to double chain, how do the cleavage angles change?

    ii. What would you expect to be the cleavage for a mineral with a wider width of modules (*e.g.*, triple chains)?

    iii. What would you expect in terms of cleavages for a mineral with an infinitely wide module, and what is this mineral?

We will now disassemble the amphibole structure back into the four *T-O-T* modules with which we started. Be sure to pull the modules straight along the direction of the widthwise pegs. **Do not twist the modules!** Also, do not attempt to disassemble the modules themselves.

1. Now let's extend the four double-chain modules lengthwise (along the c axis).

    a. What crystal habit(s) would this structure produce (hint: see photo above)?

    b. What health effects have been associated with these habits?

We'll break down the clinopyroxene structure now into the four single-chain *T-O-T* modules with which we started. Be sure to pull the modules straight along the direction of the widthwise pegs. **Do not twist the modules!** Also, do not attempt to disassemble the modules themselves.

2. Connect two single-chain modules side-by-side so that there is no vertical offset.

    a. How are pyroxenes structurally related to amphiboles?

**Questionnaire (5 min)**

**Circle the word that best describes your experience:**

I understand the modular relationships between 2:1 phyllosilicates, pyroxenes, and amphiboles **better/worse/the same** as before completing this lab.

I understand pleochroism **better/worse/the same** as before completing this lab.

I understand cleavage **better/worse/the same** as before completing this lab.

I understand extinction angles **better/worse/the same** as before completing this lab.

**Free response questions:**

What was your experience of building crystal structures with TotBlocks like (*e.g.,* were the instructions clear?)

Things in this lab that went well:

Things in this lab that were confusing:

Additional comments:

---

## Author Comment (AC3)

**Optical Mineralogy Lab:**
**Getting started with TotBlocks**

The biopyriboles: LEGO-like minerals

Last revised May 2, 2023

The building blocks of modular rock-forming minerals consist of tetrahedral ($T$) and octahedral ($O$) modules. There are two different variations of each module, reflecting the direction that these modules point. We denote these two variations with '+' and '−', with + meaning that the apices of the polyhedra are pointing upward when the widthwise pegs are pointing rightward, whereas − means that the apices of the polyhedral are pointing downward. Note that the $T$ modules are reversible, so the + and − notation is relative and not restricted to the modules themselves. For the $T$ modules, we'll use both the $T^+$ and $T^-$ variations. For the $O$ modules, we will only use the $O^+$ modules (no $O^-$ modules).

[Figure]

These modules are connected to make $T$-$O$-$T$ modules. **In this lab, we'll keep the $T$-$O$-$T$ modules together (we won't break them apart)**, but instead we'll look at how $T$-$O$-$T$ modules can be linked together widthwise.

To unlink the $T$-$O$-$T$ modules, always pull apart parallel to the pegs. Do not twist the pegs or they will break! However, if by chance a module breaks, don't worry. 3D printing is cheap; broken modules can easily be replaced. Please tell a TA if a module is broken.

**The most important takeaway of this lab is to go through the experience of building crystal structures.** The questions in this lab manual are there solely to guide you to think about the structural relationships between different minerals, as well as the fractal relationship between crystal structures and physical/optical properties.

**Lab activity**

**Mica group**

1. The mica group, structural formula $IM_3T_4O_{10}A_2$, is a 2:1 phyllosilicate with interlayer cations. The main mica-group minerals are annite-phlogopite (biotite series), $K(Fe^{2+},Mg)_3(AlSi_3O_{10})(OH)_2$, and muscovite, $KAl_2(AlSi_3O_{10})(OH)_2$.

   The red layers represent infinite sheets of vertex-sharing tetrahedra ($T$), commonly occupied by Si and Al; whereas the cyan layers represent infinite sheets of edge-sharing octahedra ($O$), commonly occupied by Mg, Fe, and Al. Black vertical pegs are used to hold the sheets together. In the space between the $T$-$O$-$T$ modules is the interlayer ($I$), where large cations such as K, Na, and Ca reside (these are not shown by TotBlocks). To build the crystal structure of the mica group, follow the instructions below:

   - Connect two sets of two double-chain-width $T$-$O$-$T$ modules such that there is no vertical offset between the modules. The adjoined $T$-$O$-$T$ modules represent the sheeted structure of the 2:1 phyllosilicates.

   - To vertically stack individual sheets, vertical pegs are inserted into the slots of the O layer to form the connection. Note: The vertical pegs can be substituted for marbles or 3D-printed spheres with a 2 cm diameter, which represent the interlayer cation site.

[Figure]

   b. Describe the cleavage of a mica (number of cleavages, angle). How does the crystal structure of the mica group dictate its cleavage? Mark the cleavage plane on the picture above.

   c. Annite (the Fe-bearing member of the biotite series) commonly displays strong pleochroism. Given the sheeted crystal structure of annite and where Fe is located within it, explain why annite is strongly pleochroic.

      d. Assuming that the substage polarizer is oriented N-S, when looking down (parallel to) the cleavage (see example photo above), is the absorption (*i.e.*, colour) stronger when the cleavage is oriented N-S or E-W?

            i. Check this on the microscope. Is this consistent with what you expected?

We will now **disassemble the mica structure** to demonstrate that the building blocks of the micas are the same as those found in the pyroxenes and amphiboles.

- Holding the mica structure sideways (one hand holding the top two *T-O-T* modules, the other holding the bottom two *T-O-T* modules), pull the two sets of modules apart to pull out the vertical pegs. **Do not twist the modules!**

- The grey vertical pegs will be stuck in one of the modules. Pull these pegs straight out of the modules.

- For each of the two sets of mica structures, hold one *T-O-T* module in each hand and **carefully** pull apart horizontally. **Do not twist the modules!** Repeat for the other set.

- You should now have four separate double-chain *T-O-T* modules and two vertical pegs. You won't need the vertical pegs anymore, so give these back to the TAs.

**Pyroxene group**

2. The pyroxenes are single-chain inosilicates with the structural formula
   **M2M1T₂O₆**. Common pyroxenes include the enstatite-ferrosilite series,
   $(Mg,Fe^{2+})_2Si_2O_6$, and diopside, $CaMgSi_2O_6$.

   Aside from the *T-O-T* modules (hosting the *T* and *M*1 sites, respectively) in the
   structure, there is another site that borders the *O* modules (the *M*2 site), which is
   slightly larger than the *M*1 site in the *O* module (not shown). We'll focus on the
   building the crystal structure of the clinopyroxenes (see figure below):

   • Connect one single-chain-width *T-O-T* module (labelled as ①) to a second
   single-chain-width module ② such that the *T⁻* module of the first ① is joined to
   the bottom of the *O* module of the second ② [and top of *O* of ① joined to *T⁺* of
   ②].

   • A third single-chain-module ③ can then be joined so that the *T⁺* of the first
   module ① is joined to the top of the *O* module of the third [and bottom of *O* of ①
   joined to the *T⁻* of ②].

   • A fourth *T-O-T* module ④ is joined to the second ② and third ③ *T-O-T*
   modules.

[Figure]

**Clinopyroxene ($C2/c$ or $P2_1/c$)**

a. Based on the assembled structure, what type of cleavage (number, angle)
   would a clinopyroxene display when viewed along the *c* axis ([001]), as
   shown on the figure above? Draw this on the figure.

b. Clinopyroxenes can be distinguished from orthopyroxenes by their
   extinction angles. The figures on the next page (top set) show the
   orientations of the principal vibration directions (X, Y, and Z) with respect
   to the crystallographic axes (*a*, *b*, and *c*) for clinopyroxenes (top) and
   orthopyroxenes (bottom), along with the orientations of the two cleavage
   planes. The bottom set of figures show the clinopyroxene and

orthopyroxene structures in side view, showing the differences in stacking and symmetry between the two pyroxene structures.

Images from pp. 205 and 209 in Introduction to Optical Mineralogy by Nesse (1991; top set) and Leung and dePolo (2022; bottom set).

ii. Based on the assembled clinopyroxene structure and the figures above, which orientation (*i.e.*, looking down *a*, *b*, or *c*) would show inclined extinction? What kinds of extinction do the other two orientations show?

iii. Why is this relevant when it comes to differentiating clinopyroxenes from orthopyroxenes? In other words, can you use any orientation to distinguish between clinopyroxenes and orthopyroxenes?

We will now disassemble the clinopyroxene structure back into the four *T-O-T* modules with which we started. Be sure to pull the modules straight along the direction of the widthwise pegs. **Do not twist the modules!** Also, do not attempt to disassemble the modules themselves.

1. Connect two single-chain modules side-by-side so that there is no vertical offset.

   a. How are pyroxenes structurally related to amphiboles?

**Amphibole supergroup**

3. The amphiboles are double-chain (ino-)silicates with the general structural formula $AB_2C_5T_8O_{22}W_2$ and include the following minerals: cummingtonite-gruenerite series, $(Mg,Fe^{2+})_2(Mg,Fe^{2+})_5Si_8O_{22}(OH)_2$; hornblende series, $Ca_2(Mg,Fe^{2+},Al)_5(Si,Al)_8O_{22}(OH)_2$; and tremolite-actinolite series, $Ca_2(Mg,Fe^{2+})_5Si_8O_{22}(OH)_2$.

   Like the pyroxenes, the crystal structure of the amphiboles contains an extra site that borders the $O$ modules called the $B$ (or $M4$) site. Like the micas, there is also a site that lies in between the $T$-$O$-$T$ modules ($A$ site). We'll focus on constructing the clinoamphibole structure (see figure below):

   - Connect one double-chain-width $T$-$O^+$-$T$ module ① to a second double-chain-width module ②, such that the $T^-$ module of ① is joined to the bottom of the $O$ module of ② (the top of $O$ for ① will also connect to the $T^+$ of ②).

   - A third double-chain module ③ can then be joined so that the $T^+$ of the first module ① is joined to the top of the $O$ module of the third ③ (the bottom of $O$ for ① should also join to the $T^-$ of ③).

   - A fourth module ④ is joined to the second ② and third ③ modules.

[Figure]

   b. Now, let's consider the cleavages of the pyroxenes, amphiboles, and micas all together. You should have the assembled structures of the pyroxenes and amphiboles in front of you. For reference, there is an extra mica structure on display at the front of the classroom. The pyroxenes have the shortest module width (single chains), followed by the amphiboles (double chains), and the micas consist of sheets.

      i. As the module width increases from single to double chain, how do the cleavage angles change?

      ii. What would you expect to be the cleavage for a mineral with a wider width of modules (e.g., triple chains)?

      iii.  What would you expect in terms of cleavages for a mineral with an infinitely wide module, and what is this mineral?

We will now disassemble the amphibole structure back into the four *T-O-T* modules with which we started. Be sure to pull the modules straight along the direction of the widthwise pegs. **Do not twist the modules!** Also, do not attempt to disassemble the modules themselves.

    1.  Now let's extend the four double-chain modules lengthwise (along the c axis).

        a.  What crystal habit(s) would this structure produce (hint: see photo above)?

        c.  What health effects have been associated with these habits?

**Questionnaire (5 min)**

**Circle the word that best describes your experience:**

I understand the modular relationships between 2:1 phyllosilicates, pyroxenes, and amphiboles **better/worse/the same** as before completing this lab.

I understand pleochroism **better/worse/the same** as before completing this lab.

I understand cleavage **better/worse/the same** as before completing this lab.

I understand extinction angles **better/worse/the same** as before completing this lab.

**Free response questions:**

What was your experience of building crystal structures with TotBlocks like (*e.g.,* were the instructions clear?)

Things in this lab that went well:

Things in this lab that were confusing:

Additional comments:

---

## Author Response (AR1)

**Author response to reviewer comments for egusphere-2023-294**

Dear Dr. Almberg,

Thank you for your editorial handling of our manuscript, "GC Insights: The crystal structures behind mineral properties – a case study of using TotBlocks in an undergraduate optical mineralogy lab" (egusphere-2023-294). In addition to the uploaded revised manuscript and supplement, as well as a track-changes document showing the changes, we provide a point-by-point reply to the reviewers' comments below. Reviewer comments are included in italic typeface. Additional author comments provided outside of the interactive discussion are highlighted in yellow.

Please review the author responses to the reviewer comments in addendum. If you have any follow-up inquiries, please do not hesitate to contact us.

Sincerely,

Derek and Paige

**Response to Reviewer 1 (Prof. David Mogk):**

Dear Prof. Mogk,

Thank you for your constructive and supportive comments on our manuscript. We appreciate your time and attention to detail. We present a response to your comments below, as well as a summary of the ways in which we intend to improve the manuscript to address these comments.

*1. This is a new and innovative way to use 3D printed materials to support and enhance student learning. Having students build their own crystal models with these 3d printed components is really cool.*

- Thank you for your praise – we hope that TotBlocks will become a more popularized tool in the future, and this reflects our commitment to keeping all design files open source, as well as publications being open access.

*2. Application of learning theory to explicitly design and implement this teaching activity is laudable. This includes active learning, constructionalism, spiral learning trajectories, and cooperative learning. This is all really great.*

*[...]*

*I'm delighted to see the application of learning theory with reference to Kolb and Fry (1975) and constructionism (Harel and Papert, 1991) (I would use constructivist theory as an alternate*

*expression, but a rose is a rose). "Spiral" learning progressions and cooperative learning approaches are also applied in this activity, and this is also supported by learning theory.*

- Thank you for your commendation. We have learned much about learning theory since starting this project, and we hope to continue learning more as the project continues.

*3. Building the 3D structures of the biopyriboles is a great way for students to understand the relationship between crystal chemistry and crystal structure. Note that these approaches have been advocated since the mid 1990s, see two references below.*

*[...]*

*Lines 20-21, In the early Teaching Mineralogy workbook that derived from a NSF-sponsored workshop followed by publication of lab exercises by MSA, two early contributions used this approach using traditional ball and stick models and also building crystal structures with templates using plasticene balls. The use of 3D printing of crystal structure components is a really nice natural evolution of this tradition.*

*Mogk, D.W., Directed-Discovery of Crystal Structures Using Ball and Stick Models, in Brady, J., Mogk, D. W., and Perkins, D., (editors), 1997, Teaching Mineralogy, a workbook published by the Mineralogical Society of America, 406 pp. available on line at https://serc.carleton.edu/NAGTWorkshops/mineralogy/activities/ballstick.html*

*Hollacher, K., Building Crystal Structure Ball Models Using Pre-Drilled Templates: Sheet Structures, Tridymite, and Cristobalite, in Brady, J., Mogk, D. W., and Perkins, D., (editors), 1997, Teaching Mineralogy, a workbook published by the Mineralogical Society of America, 406 pp. available online at https://serc.carleton.edu/NAGTWorkshops/mineralogy/activities/buildball.html*

- Thank you for sharing these references to help us better grasp the literature context in which TotBlocks sits with respect to the relationship between crystal chemistry and crystal structure. We will incorporate these references into the introductory section of the manuscript.
- These references have been incorporated into lines 19-21 of the manuscript: "Current teaching strategies for visualizing crystal structures include physical manipulatives, e.g., ball-and-stick models, paper polyhedral models, and pre-fabricated hexagonal templates (Rodenbough et al., 2015; Wood et al., 2017; He et al., 1990a; 1990b; 1994; Hollocher, 1997; Mogk, 1997)…"

*4. My main criticism is the extension of using these crystal structure models to demonstrate optical properties of minerals. Using the structures works moderately well for demonstrating the relation between crystal structure and physical properties like cleavage. However, the expression of optical properties such as pleochroism and extinction angle requires also that the orientation of principle vibration directions (X,Y,Z) are shown with respect to the crystallographic axes (a,b,c). This is not so difficult using biotite, or even ortho amphiboles and pyroxenes. But, this can be really confusing for the monoclinic varieties. I did not see that students were asked to locate the orientation of the vibration axes in or on the crystal structures. So I don't see how they can effectively determine what pleochroic colors should be*

*attributed (usually denoted as e.g., X=green Y=brown Z= red), and similarly, I'm not sure how the extinction angle can be related to the crystal model without some representation of where the vibration axes are located.*

*[...]*

*I have a little bit of an issue extending this application to optical properties as these have to be understood in terms of the relationship between the crystallographic axes (a,b,c) and the principal vibration directions (X,Y,Z). For pleochroism in biotite, this is easy to demonstrate as biotite is close to pseudo-hexagonal (monoclinic in detail), so pleochroism can easily be demonstrated parallel to the E-W or N-S cross hairs with e.g., X= brown and Y=Z dark brown. But for amphiboles and pyroxenes, this becomes more complex depending on a) is the crystal system orthorhombic or monoclinic, and b) the pleochroic color depends on which vibration direction you are looking down, and if you're not looking down one of the vibration axes you will get some intermediate color. Similarly, with extinction angle I can see why comprehension of the students did not improve as much. Again, with biotite it's simple to demonstrate parallel extinction with respect to {001} cleavage which is parallel with a vibration direction. This is also easy with the ortho amphiboles and pyroxenes where the vibration directions are parallel with crystallographic directions. But for the clino amphiboles and pyroxenes this is much more complicated for students. To be in the proper orientation to measure extinction angle, you have to be looking down the optic normal (Y) to be able to measure the correct Z to c angle. That requires understanding numerous rotational degrees of freedom to a get the crystallographic axis aligned with a cross hair, and then b) longitudinally rotate the crystal so that the optic normal is parallel with the line of sight. Any other orientation will result in an incorrect extinction angle that can range from 0 degrees (parallel) to the actual extinction angle of ~20 degrees for many clinoamphiboles or ~40 degrees for many clinopyroxenes. So, I guess the shortfall that I see here is not being able to simultaneously show the vibration directions compared to the crystallographic axes in these ToTblock models.*

*[...]*

*I don't think you can answer the question for clinopyroxene part b, which orientation would show inclined extinction without having reference to the vibration directions. And for part ii) I don't think the model as shown can address the question of what orientation is needed to demonstrate clino (inclined) from ortho (parallel) pyroxenes based on extinction angle.*

- Thank you for your constructive criticism with respect to the teaching of optical properties in our manuscript. These comments focus on the integration of principal vibration directions into the teaching of optical properties, and they concern two different optical properties: (a) pleochroism and (b) extinction angles. We will address these properties in two separate threads. The incorporation of these comments into the manuscript is addressed in the reply to Comment 5.

  o **(a) Pleochroism**

    ▪ In the case of pleochroism, the connection to principal vibration directions can be made, but it holds the potential to confuse students because of the abstract spatial nature of the optical indicatrix. On the other hand, we can use TotBlocks to predict whether absorption will be strongest along the length, width, or

height of the *T-O-T* modules, and then we can immediately verify this under the microscope. Furthermore, pleochroism is a property observed in plane-polarized light, a concept that is typically taught before cross-polarized light and the optical indicatrix, meaning that the connection between crystal structure and pleochroism should be drawn out as simply as possible, and ideally without additional knowledge prerequisites. Thus, we believe that it is better to keep the principal vibration directions out of the teaching of pleochroism for the purposes of simplicity. If desired, the relative changes in absorption can be indicated with respect to principal vibration directions, which we consider below.

- Our approach to pleochroism was rather simplistic because we only used the biotite series as an example. The goal here was to get the students to consider (1) which elements produce color in biopyriboles (in this case, transition metals such as $Fe^{2+}$), (2) where they are located in the crystal structure (in the *O* modules), and (3) how the rod-like or sheet-like configuration of the modules affects color (strong absorption occurs when the *M* sites are aligned parallel to the substage polarizer). We then (4) visually confirmed this under the microscope by projecting the microscope field of view onto an overhead screen and changing the orientation of a biotite grain observed parallel to the {001} cleavage.

- While this teaching approach uses a simple case (biotite) and does not directly consider the pleochroic formula with respect to vibration directions, it can be extended to more complex biopyribole structures and also derive the relative absorption of different principal vibration directions. For example, in the clinoamphiboles, we expect that the absorption (and intensity of color) will be strongest along the lengths of the rod modules (since the *M* sites are most coupled along this direction), then moderate along the widths, and weakest vertically, which we can verify on the microscope, using the orientation of the {110} cleavages as a frame of reference. Using the textbooks by Nesse (2012) or Deer et al. (2013), we confirm that $Z$ (~length) > $Y$ (= width) > $X$ (~height) for almost all clinoamphiboles. We can further speculate that the pleochroism in amphiboles will generally be stronger than the pleochroism in pyroxenes because the *M* sites in double chains are more structurally connected and electronically coupled than those in single chains, thus allowing for more electronic interactions with light. Thus, we can show that our simplistic case holds true without the loss of generality for other biopyriboles.

- **(b) Extinction angles**

  - With respect to extinction angles, we agree with your constructive criticism. Approaching the end of the lab, our discussion with the course professor and graduate teaching assistants indicated that it

would be more beneficial to provide a diagram showing the orientations of the crystallographic and principal vibration directions like those of Nesse (2012) or Deer et al. (2013). This would facilitate students in better understanding extinction angles. In essence, students would only need to know that the extinction angle is defined by the orientations of $Z$ and $c$, which lie in the $a$-$c$ plane. The extinction angle then becomes the angle between the cleavages (parallel to $c$) and the position at extinction (parallel to $Z$). However, we neglected to add this point as one of our learning points in the manuscript, and thus we are thankful that you have raised it.

- Regarding your comment on question 2.b.i. in the lab manual ("Based on the assembled clinopyroxene structure, which orientation (i.e., looking down $a$, $b$, or $c$) would show inclined extinction? What kinds of extinction do the other two orientations show?"), the intention of this question was to get students thinking about why orientation matters when it comes to distinguishing between orthopyroxenes and clinopyroxenes by extinction angle. Based on this lab experience, we agree that the question would be extremely challenging for an undergraduate student to answer in a one-hour lab session; however, it is theoretically possible to answer question 2.b. from first principles. For other readers in this public discussion forum, we provide a derivation that answers question 2.b. from first principles.

- **Given information:**
  - For this derivation, we know the general orientations of principal vibration directions ($X$, $Y$, and $Z$) with respect to the crystallographic axes ($a$, $b$, and $c$) for all orthorhombic and monoclinic minerals, which is based on symmetry restrictions. In orthorhombic minerals, each vibration direction must be parallel to a crystallographic direction because the crystallographic axes are all orthogonal. In monoclinic minerals, one principal vibration direction ($X$, $Y$, or $Z$) must be parallel to $b$, since $b$ is perpendicular to $a$ and $c$ (but $a$ is not perpendicular to $c$). The other two vibration directions will be at some angle to $a$ and $c$, but they will also lie on the $a$-$c$ plane because $a$ and $c$ are perpendicular to $b$ (which is parallel to one of the principal vibration directions).

  - Specific to the case of pyroxenes (but without loss of generality to other pyriboles), we know the orientations of the crystallographic axes ($a$, $b$, and $c$) with respect to the crystal structure of clinopyroxene from the figure displaying the crystal structure of clinopyroxene. We can

also determine the exact orientation of the crystallographic axis *b* on the physical TotBlocks model because it is parallel to a 2-fold rotational symmetry axis and perpendicular to a *c* glide plane (which approximates a mirror plane in the context of a second-year mineralogy class).

[Figure]

[Figure]

- We can then define *a* and *c* based on the slant height and length of the modules, resp., which is consistent with the figure. We assume that these are the same for orthopyroxenes, but with the crystallographic axis *a* representing orthogonal height rather than slant height due to the orthorhombic symmetry. From question 2.a., we know the orientation of the cleavages with respect to the crystal structure and crystallographic axes ({110} for the clinopyroxenes and {210} for the orthopyroxenes (the Miller index for *a* is doubled in the orthopyroxenes because the stacking sequence of the

orthopyroxenes doubles the length of $a$, so structurally the cleavage is still the same, despite the different indices).

- This derivation also assumes that we are only given the choice to look down $a$, $b$, or $c$ given the bounds of question 2.b.i. Considering other degrees of rotational freedom is beyond the scope of this derivation.

- **Orthopyroxenes**

  - In the orthopyroxenes, the principal vibration directions are parallel to the crystallographic axes, which correspond to the orthogonal height ($a$), width ($b$), and length ($c$) of the modules in TotBlocks. Since we know that the cleavage set of the orthopyroxenes is on the {210}, it follows that the two-cleavage section (looking along $c$) will display symmetric extinction (since the vibration directions will be parallel to $a$ and $b$, which bisect the {210} cleavage), and looking down $a$ or $b$ will both show only one cleavage displaying parallel extinction.

- **Clinopyroxenes**

  - In the clinopyroxenes, one of the principal vibration directions must be parallel to the crystallographic axis $b$, which also represents the width of the modules. The other two vibration directions will be oblique to $a$ and $c$ but they will lie on the $a$-$c$ plane because $b$ is perpendicular to both $a$ and $c$ (but $a$ is not perpendicular to $c$). The cleavage for clinopyroxenes is on the {110}.

  - On the two-cleavage section (looking along $c$), symmetric extinction will still be observed because $b$ bisects the {110} cleavage and is parallel to one of the principal vibration directions, while the other two vibration directions lie in the perpendicular $a$-$c$ plane, which also apparently bisects the {110} cleavage in this view.

  - When looking down $a$, we see one cleavage that is apparently perpendicular to $b$ (and one of the principal vibration directions) and lies in the same plane as $c$, thus resulting in parallel extinction. However, when looking down $b$ (and one of the principal vibration directions), we are looking down the normal to the $a$-$c$ plane, which allows us to see the oblique angles between the cleavage set (which is apparently parallel to $c$ in this view) and the

two principal vibration directions, thus resulting in inclined extinction.

- **Comparison between orthopyroxenes and clinopyroxenes**

  - When looking down *c*, both orthopyroxenes and clinopyroxenes display symmetric extinction. When looking down *a*, both orthopyroxenes and clinopyroxenes display parallel extinction. However, when looking down *b*, orthopyroxenes display parallel extinction, whereas clinopyroxenes display inclined extinction. Thus, the only way to distinguish between orthopyroxenes and clinopyroxenes when considering extinction character is by looking down *b*. We cannot use any random orientation to distinguish between the two – it must be looking down *b*.

- **Extension if orientation of principal vibration directions is known**

  - Although not asked in the question, we can extend the derivation to consider what other useful information we can get if the orientation of the principal vibration directions is known. For most (but not all) clinopyroxenes, $Y = b$ and thus the correct orientation for measuring extinction angles can be identified as the one-cleavage section displaying the highest birefringence (since *X* and *Z* would be visible) (Nesse, 2012).

- To summarize this subresponse, we agree that future applications of TotBlocks in teaching labs would benefit in illustrating how the principal vibration directions are oriented with respect to the crystal structure and crystallographic axes. This integration greatly simplifies the situation by representing the extinction angle ($Z\wedge c$) as the angle between the cleavage (parallel to *c*) and the position of extinction (parallel to *Z*). We present a theoretical derivation so that other readers in this public discussion forum can follow the logic behind answering such a question from first principles. However, we would like to reiterate that undergraduate students may not have the intuitive understanding of symmetry, crystal structure, and the optical indicatrix needed to present such an argument. Thus, your point about vibration directions versus extinction angles is cogent and we have incorporated these points into a modified lab manual. (File S4 in the Supplement)

*5. For publication, I think the authors need to address the relationship between crystallographic orientation and orientation of vibration directions. These would be similar to the perspective crystal drawings showing crystal form, cleavage, crystal axes, and optical orientations in standard Mineralogy texts and atlases like Deer, Howie and Zussman.  The models are really cool. Interpreting cleavage is a nice extension of the understanding of the crystal structures. But*

*I think that extension to optical properties is a step too far without additional optical orientation information.*

- Our objective in the lab activity reported in this manuscript was to help students to relate mineral structures to the properties that we can observe under the microscope, rather than specifically focusing on optical properties. We understand the confusion between these two directions, given the title in conjunction with numerous references to optical properties in the text. Our phrasing here was not precise. Thus, we intend to change the title of this manuscript to "GC Insights: The crystal structures behind mineral properties - a case study of using TotBlocks in an undergraduate optical mineralogy lab." The change in title better reflects the intention of the exercise, which is to use TotBlocks to help students link between crystal structures and the properties of minerals that are visible on the microscope (thus including cleavage), rather than focusing strictly on optical properties. We will also remove instances where we refer specifically to optical properties in the text and replace with "mineral properties" when appropriate.
- We appreciate your constructive comments regarding the integration of pleochroic formula and the principal vibration directions. However, since we hope to make direct links between pleochroism and the crystal structure of the mineral using TotBlocks, we have chosen not to add the additional complexity of these vibration directions into the teaching materials. Readers interested in how the theory of principal vibration directions intersects with optical properties can find additional explanation of these ideas in this discussion thread (which we will cite in the manuscript).
- Regarding the discussion on extinction angles, we intend to adjust the manuscript to add this learning point in Section 4. The precise language we will use in adding this point is still in flux because we await comments from a second reviewer (which may necessitate further text changes). To more fully address this point, we have also attached an updated lab manual which includes the figures from Nesse (2012) and will include this additional resource as a supplement to the revised manuscript. Due to the limitations on manuscript word count, we plan to include a note in the main text directing the audience to read this interactive discussion on the manuscript for more information with respect to the issue of vibration directions (with citeable DOI).
- Text has been added to lines 100-103 to address this point: "This gap in understanding could be addressed by communicating the role of vibration directions in understanding the optical properties of minerals. In particular, a diagram illustrating the relationship between the optical indicatrix and extinction angles might bridge the conceptual gap identified in this case study (for further discussion see Leung, 2023; File S4 in the Supplement)."

*Other comments*

*The use of manipulatives has been shown to strongly enhance learning about spatial objects—so this manuscript is well-grounded in established theory and practice. I think that this is a great exercise to demonstrate the fundamental architecture of the family of "biopyriboles" that have the same basic T-O-T layering but with single or double chains ot 2-D sheets. And it is a good way to demonstrate the physical property of cleavage, particularly for the perfect {100} cleavage of biotites. It's a bit of a jump for students to also see the {110} cleavage of pyroxenes and*

*amphiboles, but in the correct orientation looking down [001] (i.e., the "end section view")*
*students should be able to see the cleavage following the weaker bonds between the M sites at*
*~90 degrees for single chains and ~120-60 degrees for double chains. That's all great stuff.*

*For the clinopyroxene and clinoamphibole examples, question a) what type of cleavage and*
*angles,  could use some more direction as  only one cleavage direction will be seen looking*
*perpendicular to  any of the prismatic faces (parallel to c or [001]) and the two cleavages and*
*their angle will only be observed if you know to look down the c-axis or [001].  It would be a*
*better learning exercise if students were directed to look down different directions on the model*
*and then compare features.*

Thank you for your supportive comments about the relationship between manipulatives and
spatial learning and for opening discussion around the cleavages of the biopyriboles. Your
concerns with the conceptual jump required for students to see the {110} cleavage in pyroxenes
and amphiboles are well-founded and were initially shared by us. We overcame the conceptual
gap by taking advantage of the physical and modular nature of TotBlocks. When the lab was
delivered, we assembled the pyroxene and amphibole structures with more modules, and then
broke the structure apart in front of the students, showing how the modules are weaker along the
interconnections, and thus tend to break in the two planes along the {110}, leading to the 90/90
cleavages in pyroxenes and 120/60 cleavages in amphiboles. We also invited the students to
draw these cleavages on the diagram in question 2.a., giving a definite orientation for the
students to consider regarding cleavage.

We agree that it would be better to direct students to look down specific directions and compare
features. With these considerations in mind, we have included a revised lab manual, which
directs the students to look down different directions on the model and compare features. This
revised manual can serve as a more helpful resource for instructors interested in adapting the lab
for their own teaching. (This is File S4 in the Supplement.)

Thank you again for your time and care in reviewing our manuscript. Your comments helped us
think more concretely through the links we expect students to make in teaching and will improve
our revised manuscript. We hope that this discussion thread proves useful in illustrating more
concretely some of the links between crystal structure and mineral properties and that actions we
plan to take in revising the text are satisfactory to you. We are more than happy to continue
discussing these ideas with you within this discussion thread. Take good care!

References:

Nesse, W.D. (2012) Introduction to Optical Mineralogy (4th ed.). Oxford Univeristy Press,
Oxford, UK, 384 pp.

Deer, W.A., Howie, R.A., and Zussman, J. (2013) An Introduction to the Rock-Forming
Minerals (3rd ed.). The Mineralogical Society of Great Britian and Northern Ireland,  London,
UK, 549 pp.

**Response to Reviewer 2 (Prof. Brian Niece)**

Dear Prof. Niece:

Thank you for your time in reviewing our manuscript, and we greatly appreciate your constructive comments. We present a response to your comments below, and we summarize the ways in which we intend to address your comments in the manuscript.

*Let me start with some background context. I am an inorganic chemist and surface electrochemist. As a result, I came to this manuscript with an understanding of the optical properties which is perhaps only slightly more advanced than you hope the students will have at the end of the exercise. I found the illustration of how cleavage, pleochromism, and extinction angle change with structure to be clear at the level of correlating observable phenomena to structure. Deeper insight into the interaction between light and atoms is much more difficult to acquire, and I think beyond the level of this exercise.*

- We appreciate your time in reviewing this manuscript. Your comments are especially useful because they are coming from a perspective outside of mineralogy. We are grateful for your thoughts of the context and limitations of the exercise, and we are glad that the take-home message of correlating observable phenomena to structure is clear and understandable.

*As an instructor who teaches solid-state structures in my Inorganic Chemistry classes, I believe the TotBlocks will fill an important gap in the available manipulatives to help students who are just learning to think in three-dimensions develop their conceptual understanding. Models of small molecules are readily available, and models of simple crystal structures (metals and binary ionic solids) can be built from available pieces such as marbles. However, models that demonstrate how polyatomic building blocks such as the silicates can be assembled into larger lattices are lacking. The available 2D diagrams don't do justice to the structures and even rotatable computer models don't provide the tactile understanding that comes from physical models. I hope I can get a set of TotBlocks printed before the next time I teach my descriptive inorganic course. I offer here a few suggestions for improving the manuscript to make it more useful to those whose primary interest is in structure and its relation to observable properties.*

- Thank you for your supportive and encouraging comments regarding the utility of TotBlocks for helping students conceptualize three-dimensional mineral structures. If you run into any issues with 3D printing TotBlocks, please feel free to reach out to us, and we will do our best to be of assistance!

*On line 105 you note the short time spent on the exercise as a limitation. I believe this is true. This exercise could easily be expanded to fill an entire 3-hour lab session as we schedule them in chemistry courses in the U.S. In the same paragraph you note a need for "more clarity in task presentation." While I agree that additional instructions may help the students to get the structures right, I think allowing them to struggle with the assembly a bit may deepen their understanding, and a longer lab session would allow that. It would also allow more time for students to investigate the various ways the building blocks can be combined into successively larger modules from single chains to double and eventually sheets. See my note at the end for another comment on this.*

- Thank you for your constructive comment around the timing of the exercise. For background context, Derek was a graduate teaching assistant for the Optical

Mineralogy class where TotBlocks was trialled, and only an hour was allotted in the lab syllabus. We hoped to see if TotBlocks would be useful for further integration in future classes. The time limitation jumped out to us when reflecting on the results of this trial exercise, and we thought more careful instruction would help with communicating the structure, but your observation that allowing the students to wrestle with assembly and experiment with different combinations is really cogent. One way that students could grow from the exercise is the 'discovery' of other mineral structures. Even in our short lab session, we observed several groups initially constructing a palygorskite linkage rather than an amphibole linkage, and that experimentation allowed us to further develop the idea of these molecular building blocks in conversation with them.

- In the future, we hope to not only use TotBlocks in a full lab session, but to extend the use of them throughout a course to support learning different mineral structures during the term.

*On lines 116-117 you note that you do not know whether a student's learning experince would be better or worse without the TotBlocks. I would note that your have only measured their self-reported change in understanding. You have not reported an external measurement of improved learning (which, as you note, would have had a control). I don't think you need to do anything else here. I am convinced of the utility of the blocks. Just be careful in your phrasing to make sure you acknowledge the limits of your data.*

- Thank you for your affirmation of the utility of TotBlocks. It means a lot coming from an experienced educator (particularly an inorganic chemist!). This comment was really helpful to us because reflecting on the control group was repeatedly discussed between Paige and Derek. We are hyperaware of the challenges of not having an external measurement of improved learning, and at the time of submitting the manuscript, we agreed to take a cautious and self-critical approach to our data. We fully agree with your comment and appreciate the language you have used here which helps us better communicate our data limitations. We plan to change lines 116-117 of the manuscript to read along the lines of "Finally, this study relies on self-reported reflections and lacks an independent metric for assessing learning improvement (i.e. a control group)."

- Lines 115-116 have been changed to read, "Finally, this study relies on self-reported reflections and lacks an independent metric for assessing learning improvement (i.e. a control group)."

*In the lab handout, on p. 2, you note that the interlayer cations are not shown by TotBlocks. They could be! It would be easy to incorporate loose spheres between the layers to demonstrate occupancy of those positions. In addition to illustrating where the cations go, the form of these models would make clear how they are more loosely bound, and why those positions are not always fully occupied (or occupied by identical ions). A short comment about what size marbles or 3D-printed spheres would be suitable would help instructors who want to add this to the lesson.*

- We totally agree that including interlayer cations in the mica structure is feasible in TotBlocks and think that the loose spheres you envision would work well in the short term.

- A sphere of roughly 1.2 cm diameter would be a good proxy for a $K^+$ ion in [12] coordination (the interlayer cation of muscovite). However, as the site involves K–O bonds in [12] coordination, the size of the interlayer site is larger and can be roughly represented by a sphere with a diameter of ~ 2 cm (accounting for the extra vertical padding given to the *T* modules). We have updated the TotBlocks GitHub instructions (https://derekdvleung.github.io/totblocks/) for constructing the mica structure with this information for instructors who are keen to include interlayer cations in their lessons.

- In the longer term, we plan to design some hexagonal prisms to represent the shape of the [12]-coordinated interlayer-cation site, which would occupy the space of the site more precisely (and allow the modules to stack more easily) and help students more precisely visualize the coordination of the ions. This is planned in the next iteration of TotBlocks.

*On p. 3, you note that clinopyroxenes can be distinguished from orthopyroxenes. It doesn't look like you have them build both, and the distinction is not clear in the figure. The right half of the new figure you added in your response to the first review is very helpful in this regard, and should be incorporated. In addition, I believe having the students build both would help their understanding of the layers stack differently in the two structures.*

- Good insight! We have incorporated the figure from Leung and dePolo (2022) into the revised lab manual (see attached file). (In the manuscript, this file is referred to as File S4 in the Supplement.)

- When designing the lab presented in this case study, we were worried that trying to incorporate too many crystal structures in the limited time of the lab session would result in the students not having adequate time to reflect on the structures themselves. We did not have them construct orthopyroxene to try to balance the lab activities.

- However, you're exactly right that having the students build both structures would allow them to more clearly understand the stacking differences between these polytypes. Ideally, we think the best instructional solution would be a separate 'pyroxene' lab exercise focused exclusively on building the orthopyroxene structure and the clinopyroxene structure, so that the differences in optical properties resulting from these symmetry differences can be more concretely discussed.

*It seems that you have the students dissasemble the mica structure along the cleavage plane, but not the other two models. If it can be done without damage to the models, it would be instructive to have them try. It looks as if grasping the top/right pair of modules in the clinopyroxene and the amphibole and pulling to the right while holding the bottom/left pair would cause cleavage along the correct plane, while pulling in other directions would not.*

- Great point! You have a keen eye for visualizing these cleavage planes. During the instruction portion of the lab, these cleavages were demonstrated to the class as a whole in the exact way you've described (see our response to the first of Prof. Mogk's 'Other Comments' in RC1). We then invited the students to draw the cleavage planes for pyroxene and amphibole on the figures provided in the lab manual. Inviting the students to manipulate the structures themselves and further test their understanding of this concept is a great thought. We have amended the text of the revised lab manual to

include this invitation to the student. This can be done and will be incorporated into a future lab.

*Here are my responses to the review criteria:*

1. *Does the paper address relevant scientific questions within the scope of GC? Yes. The techniques presented are likely to lead to improved student understanding of mineral structures.*

2. *Does the paper present novel concepts, ideas, tools, or data? Yes. As far as I know TotBlocks are unique in allowing easy construction of models of this type of mineral.*

3. *Are the scientific methods and assumptions valid and clearly outlined? Yes.*

4. *Are the results sufficient to support the interpretations and conclusions? Yes. See my note above about being clear that learning is student-reported.*

5. *Do the authors give proper credit to related work and clearly indicate their own new/original contribution? Yes.*

6. *Does the title clearly reflect the contents of the paper? Yes, particularly with the change already suggested in response to other reviews.*

7. *Does the abstract provide a concise and complete summary? Yes.*

8. *Is the overall presentation well structured and clear? I found the manuscript to be quite readable, and the lab handout should be easy for students to follow.*

9. *Is the language fluent and precise? Yes.*

10. *Are the number and quality of references appropriate? Yes. The authors place their work carefully in the context of the pedagogical literature.*

*I recommend publication with minor revision.*

*-Brian*

- Thank you for providing your responses to the review criteria to us and for your supportive assessment of the manuscript! We really appreciate your thoughts in helping us firm up the fine details of this work.

*Final note: As a chemist, I am interested in structure and bonding beginning at the level of individual atoms. The tetrahedra and octahedra can easily be built from available modeling kits. Models designed to bridge the gap between those and the ToTBlocks would be a welcome addition. That is, tetrahedra and octahedra that can be assembled into dimers, trimers, rings, and ultimately the single- and double-chains that are your smallest pieces. I have designed enough 3D models to know how much work that would involve. Since this is not the paper in which you introduce the models themselves, it would not be appopriate to add here. But I believe they would be a valuable addition to what you have produced.*

- Thank you for your suggestion (and for your understanding in seeing that incorporating these models of individual tetrahedra/octahedra is outside of the scope of this particular paper). I (Derek) had considered making the individual tetrahedra but was more focused on making the mineral structures (and *T-O-T* modules made sense as the functional unit in that case). But the transition from individual molecules to these basic chain structures is an important conceptual link for students to grasp. Creating

individual tetrahedra should be relatively straightforward, but the octahedral models would present more of a challenge because it requires more thought to design widthwise pegs that are compatible with the vertical pegs from the $T$ modules. This design challenge is certainly not insurmountable, and your comment has renewed my motivation to explore this design more concretely for future iterations and expansions of TotBlocks.

Thank you again for your helpful and constructive comments for our manuscript, which have undoubtedly refined the manuscript for future publication. We appreciate your time and effort in reviewing our manuscript!

Sincerely,

Derek and Paige